# Healthy Foods and Healthy Diets. How Government Policies Can Steer Food Reformulation

**DOI:** 10.3390/nu12071992

**Published:** 2020-07-04

**Authors:** Mathilde Gressier, Franco Sassi, Gary Frost

**Affiliations:** 1Section for Nutrition Research, Department of Metabolism, Digestion and Reproduction, Faculty of Medicine, Imperial College London, London W12 ONN, UK; g.frost@imperial.ac.uk; 2Center for Health Economics & Policy Innovation, Department of Economics & Public Policy, Imperial College Business School, Imperial College London, London SW7 2AZ, UK; f.sassi@imperial.ac.uk

**Keywords:** public health policy, food reformulation, food policy, policy design

## Abstract

Food reformulation policies aimed at reducing the risk of diet-related non-communicable diseases have been implemented in many countries. The degree of success of reformulation policies in changing the range of food options available to consumers has been a function of the design of these policies. Our objective was to review the different factors making the design and implementation of a food reformulation policy effective at improving populations’ diets and health. In this narrative review, we present a logic model of the action of reformulation policies on consumer behaviour, dietary intake and population health. We set out how policy design could drive outcomes, and highlight the role for governments and public health agencies in promoting food reformulation that is effective in improving diet and health. The key drivers of success for reformulation policies include strong incentives, a tight implementation strategy, a focus on the overall nutritional quality of food products, rather than on individual nutrients, and effective monitoring and evaluation. Additionally, policies should mark the distinction between product reformulation and product differentiation, which have different nutrition and health outcomes.

## 1. Introduction

Twenty-two percent of global deaths amongst adults are attributed to diet-related risks [1]. Many countries have adopted population-wide policies aimed at improving diets and countering a steady rise of obesity and non-communicable diseases, such as type 2 diabetes, cardiovascular disease and cancer. Current diets worldwide are characterised by an increasing intake of manufactured foods, defined by some level of alteration during their production (e.g., heated, refined, milled, or mixed with other ingredients). Packaged foods account for 61% of retail sales in the United Kingdom, 42% in Thailand and 32% in Brazil [2]. Manufacturers constantly review and adjust the formulation of the foods they produce to adapt them to consumer preferences and demand, to lower production costs and increase profits, to better position their products relative to those offered by competitors, or in response to government regulation or guidelines. While food reformulation is and will remain primarily driven by market forces, the idea that reformulation efforts can be steered in the direction of an improved nutritional quality has been gradually developing in the public health policy debate [3,4]. Food reformulation has the potential for generating major public health impacts, but realising this potential will require greater clarity and agreement on the underlying concepts and mechanisms through which reformulation can have an impact on people’s diet and health.

## 2. Food Reformulation Policies

A key aim of policies designed to incentivise food reformulation is to improve the nutrient composition of foods available to consumers while not detracting from other product characteristics that are known to drive consumer choices, such as taste, convenience, or affordability [5]. Traditionally, fortification strategies were used to improve public health by addressing common nutrient deficiencies or by adding back nutrients lost in the manufacturing process, such as vitamins and minerals in bread and breakfast cereal [6]. With the increase in non-communicable diseases (NCDs), many governments have adopted reformulation policies to protect the population against these NCDs, focusing mostly on the reduction in sodium or trans-fatty acids (TFA) in foods. More recently, the focus of reformulation policies has shifted to added sugar and energy reduction [7], and sparse initiatives were launched to increase the nutrient density of foods by increasing their amount of fibre, whole grains, or specific fats such as omega-3 [8,9]. The debate on the value of reformulation has also widened in scope, extending to characteristics of foods other than the nutrients they contain, including changes in structure, texture, or additives which could make foods, and people’s eating habits, healthier [4,10]. However, in the context of policies to promote food reformulation as a public health strategy, changing the nutrient profile of food products to reduce the risk of diet-related non-communicable diseases remains the central goal.

## 3. Review Aims and Methodology

This review aims to assess how the design of food reformulation policies has an impact on their effectiveness. We aimed to identify the pathway through which reformulation policies can have an effect on consumer’s health. We reviewed published evidence to inform policy makers and researchers about how to best design food reformulation policies. We searched the published and grey literature using Embase, Global Health and Medline databases. We searched for literature reviews and the studies they included.

## 4. Reformulation, Derived Food Products and Consumer Choice

Food manufacturers often respond to government regulation aimed at changing nutrient intakes by launching new products, as well as by reformulating existing products [7,9,11]. However, these two strategies have different impacts on consumer behaviour. The reformulation of existing food products (e.g., a gradual reduction in sodium) does not change the range of choice options available to consumers, as reformulated products replace previous versions. Conversely, introducing new versions of existing products, with improved nutrient composition, extends the range of choice options, leaving both the old and the new versions on the market (e.g., the introduction of a sugar-free drink while retaining the sugar-sweetened version). The success of the latter strategy in making consumers’ diets healthier has to rely on the willingness of individual consumers to switch to the nutritionally balanced option. When the original products remain available, consumers may not change their food choices because of inertia, or to avoid the ‘taste cost’ of changing habits [12]. Ways of improving the nutrition composition of food products can be viewed as a function of a gradient of consumer involvement needed for them to benefit from the improved product (Figure 1).

## 5. How Can Governments Incentivize Food Reformulation?

The options for governments to incentivize reformulation lie along a continuum between market approaches in which consumer demand is the only driving force, to mandatory limits imposed by governments [13].

### 5.1. Helping to Create a Healthier Food Supply

Reformulation is often an expensive and uncertain endeavour for manufacturers, worth pursuing only when there is a realistic expectation of increased sales and margins relative to the status quo. From a manufacturer’s perspective, typical drivers of reformulation include an increased demand for the reformulated product from consumers, and competitive or regulatory threats to their business. Technical support in the development of improved products could encourage reformulation, especially for smaller manufacturers with reduced research and innovation resources and capabilities. Mandatory or voluntary reformulation targets can also promote reformulation. Mandatory targets, and even outright bans, have been successfully implemented for TFA, for which strong evidence of detrimental health effects may have contributed to a wide acceptance of the policy [14].

Public–private partnerships (PPPs) are agreements between governments and manufacturers involving the joint setting of reformulation objectives, which has made this approach an attractive option for manufacturers. However, some PPPs lacking a strong institutional base have not succeeded in creating sufficient incentives for reformulation [11]. PPPs require careful design and monitoring in order to be as effective as tighter command and control approaches in incentivizing reformulation. Laverty et al. showed that the transition from a tightly controlled to a more lenient salt reduction strategy in the UK led to smaller improvements in population salt intake [15].

### 5.2. Helping to Create a Demand for Nutritionally Balanced Products

Governments can also promote reformulation by steering consumer demand towards nutritionally balanced foods. By informing consumers about the health benefits and risks linked to certain foods, or nudging them through environmental cues, government policies can influence the demand for certain categories of food products. For example, the introduction in the United States of a new dietary guideline on whole grain consumption pushed manufacturers to develop food products using whole instead of refined grains [9]. In addition, nutrition labelling aims to guide consumers into making healthier choices. Labelling policies—both mandatory nutrition labelling and voluntary front-of-pack logos—have been shown to provide incentives for manufacturers to reformulate their products [16]. Manufacturers anticipate a likely shift in consumer demand towards nutritionally balanced products, to coincide with a wider use of nutrition labels. Labelling has been shown to improve the quality of purchases in experimental settings [17,18]. In real shopping conditions, the improvement was smaller and came mainly from switches between products of the same category [19]. Labelling and dietary guidelines have a two-fold potential to improve populations’ intakes—they inform consumers about how to choose nutritionally balanced products, and they incentivize reformulation.

### 5.3. Interplay between Supply and Demand

The definition of the reformulation target, either coming from a demand or a supply approach, determine the nature of the changes in the food environment. While policies requesting gradual changes may promote reformulation, policies requesting larger changes (such as taxation with only one threshold) may favour to the introduction of new products in the market. In both cases, the aim is that consumers choose nutritionally balanced products. To that end, governments can make these products cheaper than the original version (e.g., when sugar-sweetened but not artificially sweetened beverages are taxed). Improved products can also be promoted with a nutrition claim (such as “reduced in salt”), or a logo (such as a color-coded front-of-pack labelling). However, educating populations to make healthy choices has been shown to be costly and requires a permanent input [20], while the ability of front-of-pack labelling to help consumers making healthier choices in real conditions is not clearly established [16,21]. In addition, some research has suggested that claims or logos can favour overconsumption by creating a “halo effect”: when the food appears healthier, consumers can respond by consuming more of the product [22]. Overall, changes in supply and in demand seem to be needed to see improvement in dietary intakes, as systematic reviews on salt and TFA reduction strategies showed that interventions with multiple components, including structural changes such as reformulation, and educational components such as campaigns or labelling were the most effective in improving populations’ intakes [14,23].

## 6. How Could Food Reformulation Policies Improve Population Health?

Food reformulation policies can be a means of improving diet quality and reducing diet-related diseases (Figure 2). The logic model in Figure 2 was designed using published evidence on the impact of reformulation on the food environment, consumer behaviour and dietary intakes. The aim of this logic model is to show the different steps through which food reformulation has an impact on consumer health. Reformulation policies first change the food environment to induce better food choices to consumers. For example, the TFA restriction in New York City led to a reduction in the amount of TFA in restaurant foods [24]. When a nutritionally balanced reformulated product replaces a previous version, consumers do not have to make a choice to benefit from the improved nutritional profile of the product that becomes the default choice. Experimental studies showed that a gradual reduction in sodium was not noticed by consumers, contrarily to an abrupt reduction [25]. Similarly, a substantial reduction in the added sugar content of foods and beverages was noticed by individuals and led to energy compensation [26]. Gradual reformulation, opposed to larger changes in food composition, would be expected to contribute to improving dietary intakes so long as the population does not compensate for the change in nutrient intake. In experimental settings, the reduction in sodium or saturated fatty acids in foods led to reduced intakes of these nutrients, suggesting no compensation [27,28]. Governments can learn, from the different steps of the logic model and findings from experiments and implemented reformulation policies, how to frame reformulation policies so as to make sure the changes will improve the population’s health.

## 7. Limiting Detrimental Choices and Compensation

As reformulation relies on small and gradual improvements, the more food products are reformulated, the bigger the beneficial impact on health. For example, the limited impact of Australia’s Food and Health Dialogue has been linked to the fact that many products were not subject to the sodium reduction targets [11]. Therefore, changes in the food environment were not sufficient to change dietary intakes. Reformulating the majority of products reduces the risk that consumers would switch to a non-reformulated product and ensures that the whole population benefits from reformulation. However, systematic reviews often noted disparities between manufacturers in the extent by which they reformulated the same type of product. Often, a great variability in the content of a nutrient remains amongst similar products, suggesting that some but not all products were reformulated [11,29]. The reformulation of only a subset of products (e.g., expensive products) could create disparities in population health, where only rich or health-conscious consumers benefit from improved products [30]. This regressive effect of reformulation was noted in Denmark, where the voluntary TFA reduction left some people with high TFA intakes, as some foods with high level of TFA remained in the market [13]; this was tackled by the establishment of a total ban on TFA, including for restaurant foods, and imported products.

## 8. Ensuring an Improvement in Food Composition

In addition, for reformulation to have a positive impact on diets or health, the quality of food has to be improved as a whole. Most reformulation policies launched so far had a single nutrient focus (e.g., sodium or TFA). Their evaluation focused on the targeted nutrient, often ignoring the complete nutrient profile of food products. However, research suggests that reformulation strategies focusing on a single nutrient were sometimes accompanied with a negative change in another nutrient (i.e., in the opposite direction than a reformulation to improve public health would have expected) [7,31]. For example, out of the top 20 brands that reduced their added sugar content as part of the UK sugar reduction programme, 6% increased both SFA and energy, and 45% increased one of these two nutrients [7]. In addition, reformulation may lead to products with more additives to counterbalance the loss of an ingredient with specific characteristics. While the causes remain unknown, there is growing evidence linking the consumption of ultra-processed foods with an increased mortality, possibly because of some additives [10]. The positive effect of reformulation may be compromised if food quality is not considered in its entirety.

## 9. Challenges to Monitor the Effect of Food Reformulation Policies

Previous research on reformulation showed that while many policies were launched, there were only a few evaluations of their impact on population health [32]. Thus, their relevance for improving public health remains uncertain. Several factors make the evaluation of reformulation policies challenging. Firstly, reformulation, as a population-wide intervention, is difficult to evaluate given the multiplicity of other factors confounding its effects. Reformulation policies are often embedded in wider public health initiatives comprising other components [14,23], making it difficult to disentangle the effect of each component. New methods were recently used to disentangle the proper effect of reformulation from consumer behaviour changes and market renewal [33,34]. However, these studies could not identify the factors explaining why consumer behaviour changed (possibly because of changed advertising or public health campaign).

A rigorous evaluation of food reformulation policies is necessary to understand each step of the logic model described in Figure 2. Additionally, monitoring policies are a key component to motivate manufacturers to voluntarily reformulate their products [3]. For that, different data must be collected at baseline, and after the implementation of a reformulation policy, namely product-level information, purchase patterns, dietary intakes and resulting health outcomes. Progressing on the logic model from the food system towards health impact increases the risk of confounders. An understanding on all steps of the logic model is needed to give policy makers insight into how best to design a reformulation policy that will be effective in improving health.

## 10. Discussion and Conclusions

Given the large contribution of manufactured foods in the diet in many industrialized countries, policies aimed at improving their composition can have a broad public health impact. However, the mechanism behind this impact has been understudied. In this paper, we propose a logic model to guide policy makers and researchers to design effective policies and evaluate them. Although the logic model has been mostly built using evidence from salt and TFA reformulation, it should apply to all types of reformulation, and can be used as a framework while assessing evidence on the effect of reformulation. The principles it relies on (silent and gradual reformulation, inducing no compensation from consumers) are expected to be valid for all types of reformulation. However, further research is needed to test the validity of some steps. In particular, the assumption that reformulation does not provoke a change in behaviour is the cornerstone of this logic model, but only few studies evaluated consumer’s reaction to reformulation in real settings. Although most evidence used in this paper is from developed countries, the same principles should apply to low- and middle-income countries that also derive an important part of their energy from manufactured foods. Policy makers should learn from successful policies implemented in other countries; these policies show the feasibility of reformulation and successful policy design.

In line with past successful examples of food reformulation policies implemented, reformulation should be part of wider initiatives informing consumers about healthy choices and dietary patterns, and improving other aspects of the food environment such as food availability. The careful design of reformulation policies and the monitoring of their implementation are of utmost importance to guarantee that these policies lead to improved population health.

## Figures and Tables

**Figure 1 nutrients-12-01992-f001:**
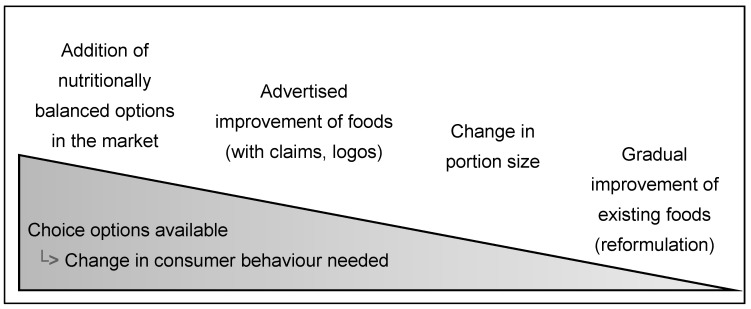
Classifications of strategies changing the characteristics of food available for a population along a gradient of change in consumer behaviour needed to get a benefit from the new food products.

**Figure 2 nutrients-12-01992-f002:**
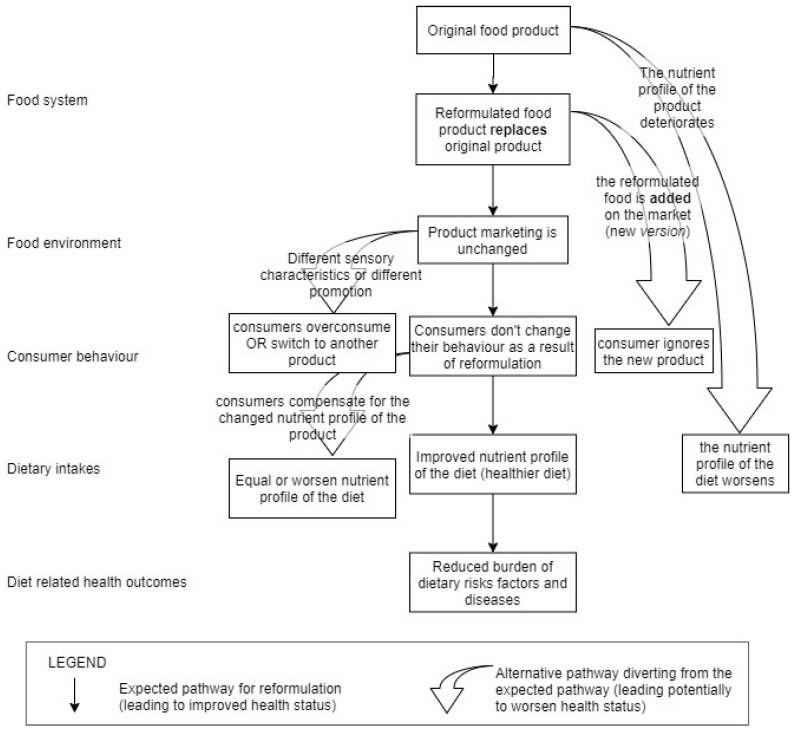
Logic model of the effect of reformulating existing food products, and sources of diversion from the expected health benefits.

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
