# Peer review of "Healthy Foods and Healthy Diets. How Government Policies Can Steer Food Reformulation"

_nutrients, 2020, doi:10.3390/nu12071992_

Round 1
Reviewer 1 Report
Introduction:
Considering that the term healthier has several interpretations, may consider defining what healthier means in the context of this manuscript or else changing to nutritious or nutrient dense.
Line 29: recommend clarifying with ‘manufactured processed foods’ as frozen vegetables is considered manufactured, but more nutrient dense compared to crackers. Also, mentioned in the beginning of the sentence worldwide, yet only provided a stat from the UK. Recommend rewording this if the focus will only be on the UK for that statement.
Line 45: clarify to added sugars
Even though it was mentioned that education is expensive, it is a necessity. Public health campaigns have been out focusing on reducing certain nutrients, exercising more, etc, yet the focus is so broad that many people do not believe it pertains to them. Therefore, there should be an elaboration on this concept and how to create low-cost targeted educational campaigns to enhance the shift towards consuming more nutritious products.
In this manuscript, the polices were highlighted based on developed/higher-income countries, but no discussion about developing or lower-income countries where there is a nutrition paradox. In which several individuals living in those countries have NCDs at a rate close to or greater than other developed countries, which may be due to lack of policies in these countries, so there should be mention of this.
Reviewer 2 Report
This paper synthesizes various aspects of food reformulation policies. The overall objective and methods of the study are unclear. Specific comments are listed below.
Abstract:
- The authors ought to clearly state the nature of their study (type of review?) and make the objectives of the review clear. This remark applies also to the main text.
- “We assess how policy design can drive outcomes…” – this is not correct as there is absolutely no assessment of the effect of policy design on outcomes, no data collection/analysis, etc.
- “Key drivers of success for reformulation policies include…” – missing is the actual motives for reformulation and the mandatory/voluntary nature of the target.
Lines 30-32: “Manufacturers constantly review and adjust the formulation of the foods they produce to adapt them to consumer preferences and demand, to lower production costs, to better position their products relative to those offered by competitors, or in response to government regulation” – missing is the motivation “to maximize profits” as well as motivations in response to updated dietary guidelines.
Lines 42-44: “Many governments have adopted policies to promote reformulation, focusing mostly on the reduction of...”- missing are policies regarding enriching foods with vitamins, omega-3, etc. Also policies regarding additive reduction should be more clearly stated.
Figure 2: How was this theoretical model developed? The authors’ intuition? Did the authors use any existing models in this domain? What data sources were used, if any? The authors are urged to integrate information regarding the empirical support for the various links in order to clearly/visibly distinguish the associations with strong empirical support (ie, those that have been tested and established) versus those that are purely theoretical?
